# Is Intracanal Cryotherapy Effective in Reducing Postoperative Endodontic Pain? An Updated Systematic Review and Meta-Analysis of Randomized Clinical Trials

**DOI:** 10.3390/ijerph182211750

**Published:** 2021-11-09

**Authors:** Amal Almohaimede, Ebtissam Al-Madi

**Affiliations:** Department of Restorative Dental Sciences, Endodontic Division, College of Dentistry, King Saud University, Riyadh 11451, Saudi Arabia; ealmadi@ksu.edu.sa

**Keywords:** cold therapy, cryotherapy, endodontic pain, intracanal cryotherapy

## Abstract

This research aimed to assess the potency of intracanal cold therapy in diminishing postoperative endodontic pain. PubMed, Scopus, the Cochrane Library, EMBASE, the Web of Science, grey literature, and endodontic journals were used to identify randomized controlled clinical trials evaluating postoperative pain after a final irrigation with a cold irrigant (as an experimental group) and a room temperature irrigant (as a control group). The risk of bias was rated according to the Cochrane Collaboration’s tool and the Grading Recommendation Assessment, Development, and Evaluation (GRADE) system was used to estimate the evidence quality. For the meta-analysis, a random effects model was utilized. The qualitative analysis contained 16 studies and the quantitative analysis contained 9 studies. The experimental groups showed a reduction in postoperative pain at 6 h (mean difference (MD) = −1.11; *p* = 0.0004; I^2^ = 72%; low quality evidence), 24 h (MD = −1.08; *p* = 0.003; I^2^ = 92%; low quality evidence), 48 h (MD = −0.38; *p* = 0.04; I^2^ = 81%; low quality evidence), and 72 h (MD = −0.69; *p* = 0.04; I^2^ = 90%; low quality evidence). A higher quality of evidence from more clinical trials is needed.

## 1. Introduction

One of the essential parts of endodontic therapy is to prevent and manage postendodontic pain [1]. The prevalence of postoperative pain after endodontic treatment ranges between 3 and 58% [2]. The reported frequency of teeth that showed persistent pain at 6 months or more after root canal treatment varied from 4.9–12% [3,4,5,6,7]. Moreover, the prevalence and severity of pain were not shown to differ significantly among the number of root canal treatment visits [8]. Postoperative pain following endodontic treatment is due to chemical, mechanical, or microbial injury to the periradicular tissue [9]. Several techniques and treatments have been described in the literature to control postoperative pain in endodontics including the prescription of medication [10,11,12], the use of intracanal therapies [13,14,15], and occlusal reduction [16]. Nevertheless, each technique and treatment has its own disadvantages. Nonsteroidal anti-inflammatory drugs (NSAIDs) have been documented to have harmful effects on several body systems such as the gastric mucosa and hepatic system [17]. Furthermore, the initial exposure to opioids might lead to respiratory depression, nausea, and the risk of long-term use, abuse, and overdose [18]. The use of intracanal therapies (such as calcium hydroxide, laser application, and analgesic solutions) and occlusal reduction have contradictory effects on reducing postoperative endodontic pain in the literature [19,20]. Cryotherapy involves reducing the tissue temperature for curative purposes [21]. It was first used by the ancient Egyptians circa 3000 BCE to cure injuries and decrease inflammation [22]. Since 1960, it has been used in medicine to relieve the pain of sports injuries [23]. Its mechanism of action in reducing pain involves decreasing the tissue temperature and reducing the flow of blood and metabolic activity [24]. In dental practice, cryotherapy reduces edema and postoperative pain following third molar extractions [25]. In endodontics, cryotherapy was introduced in several randomized clinical trials (RCTs) as an intracanal irrigation method to reduce postoperative endodontic pain [26,27,28,29,30,31,32,33,34,35,36,37,38,39,40,41,42,43,44,45,46,47,48,49,50,51,52,53,54,55,56,57,58,59,60,61,62]. However, the evidence from these RCTs has been debatable; accordingly, this systematic review and meta-analysis aimed to assess the potency of intracanal cryotherapy in diminishing postoperative endodontic pain.

## 2. Materials and Methods

### 2.1. Study Protocol

This research was recorded with the International Prospective Register of Systematic Review (PROSPERO CRD42021253682). The recommendations of the Preferred Reporting Items of Systematic Reviews and Meta-Analysis (PRISMA) statement were followed by the authors [63].

### 2.2. PICOS Eligibility Criteria

Participants (P): Adult patients (≥18 years old) with permanent teeth that had pulpal or periradicular pathosis and who were undergoing nonsurgical root canal therapy.

Intervention (I): A cold irrigant was used as the final irrigant.

Comparator (C): A room temperature irrigant was used as the final irrigant.

Outcome (O): Postoperative pain level from day one.

Study design (S): Randomized controlled clinical trials evaluating postoperative pain after instrumentation or obturation.

### 2.3. Database Sources and Search Strategy

The main electronic databases were utilized to identify the studies: MEDLINE through PubMed (1952–June 2021); Scopus (2004–June 2021); EMBASE (1984–June 2021); the Cochrane Library (1993–June 2021); and the Web of Science (1997–June 2021). In addition, a grey literature search was carried out on OpenGrey and Google Scholar. Moreover, endodontic specific journals such as the Australian Endodontic Journal, the International Endodontic Journal, the Journal of Endodontics, Clinical Oral Investigations, and Endodontics & Dental Traumatology were searched for relevant articles. The lists of references of the selected studies were manually searched for any relevant documented articles.

On the basis of the PICOS format, the eligible studies were included by defining the MeSH (Medical Subject Headings) terms, keywords, and synonyms in the search strategy. The search terms were “cryotherapy”, “intracanal cryotherapy”, “cold therapy”, “endodontic pain”, “postoperative endodontic pain”, and “endodontic cryotherapy”. Studies published in the English language were included with no restrictions on the publication date. Case reports, review articles, nonhuman studies, and retreatment cases were excluded. Duplicate articles present in more than one database were considered to be only one article.

### 2.4. Selection of Studies and Data Collection

The titles and abstracts were independently screened by two evaluators (A.A. and E.A.-M.). They judged the study eligibility, rated the risk of bias, and examined the reported data [64]. If data applicable to the inclusion criteria were not found in the abstract or the abstract was missing, then the article was accessed for full-text reading. Any controversy between the reviewers was discussed. The data extraction was conducted using an Excel spreadsheet (version 14.7.1, Microsoft, Redmond, WA, USA). The retrieved data included the year of publication and the authors of the study, the study design, the pulpal and periapical diagnosis, the age range of participants, the type of teeth, preoperative pain in the experimental and control groups, the postoperative drug prescription, the pain evaluation scale utilized, the total sample size, the irrigation methods utilized, the type and concentration of irrigant utilized, the irrigant activation, and postoperative pain one day minimum after the procedure, the results, and the conclusion. The risk of bias was rated according to Cochrane Collaboration’s tool for assessing the risk of bias in randomized trials (RoB 2) [65]. For the meta-analysis, RevMan software (version 5.4; The Cochrane Collaboration, London, UK) was used. To indicate the effect estimate, the mean differences (MDs) with 95% confidence intervals (CIs) and a random effects model were used. The I^2^ index was used to test the heterogeneity. The quality of evidence was judged using the GRADE system (www.GradeWorking-Group.org, accessed on the 1 September 2021) [66].

## 3. Results

A total of 33 studies remained for retrieval and an eligibility assessment after excluding ineligible and duplicated articles (Figure 1).

Seventeen articles were eliminated due to the following causes: ten studies were still ongoing [26,27,28,29,30,31,32,33,34,35], two studies were in vitro [36,37], two studies were critical appraisals [38,39], one paper was a letter to the editor [40], one was an editorial paper [41], and one study had no control group [42]. Therefore, 16 studies were involved in the systematic review [43,44,45,46,47,48,49,50,51,52,53,54,55,56,57,58]. Seven of the sixteen studies documented their results as percentages, which were excluded from the meta-analysis [43,44,47,50,52,55,56]. Nine studies involving 878 participants were included in the meta-analysis [45,46,48,49,51,53,54,57,58]. The main features of the sixteen included articles are summarized in Table 1.

Eight of the sixteen articles consisted of two parallel groups [44,45,47,48,50,53,54,57], four studies had three parallel groups [43,49,51,58], three studies had four parallel groups [46,52,56], and one study had five parallel groups [55]. The number of participants in these studies ranged from 30 to 240 with ages ranging from 18 to 70 years diagnosed with irreversible pulpitis in 11 studies [43,44,45,46,47,49,50,52,53,55,57], necrotic pulp in 3 studies [48,54,58], and pulpal and periradicular pathosis or normal periradicular tissues in 12 studies [44,45,46,47,48,49,53,54,55,56,57,58]. The assessment of postoperative pain ranged from day 1 to day 7 and it was measured using a visual analogue scale (VAS, 0–10 cm) in 12 studies [43,44,45,47,48,49,50,51,52,53,54,56]. Three studies used the VAS (0–100) [46,55,58], one study used the Heft–Parker scale [57], and all of the scales were converted into a 0–10 cm VAS for statistical calculation standardization. Preoperative pain was evaluated in 13 studies [43,44,45,46,47,48,49,51,53,54,55,56,57] ranging from no pain in 2 studies [49,51] to mild pain in 2 studies [43,44], moderate pain in 2 studies [53,54], and severe pain in 6 studies [45,46,47,48,55,56]. One study had fourteen patients with mild pain, twelve patients with moderate pain and fourteen patients with severe pain [57]. Three studies did not clearly describe the preoperative pain assessment [50,52,58]. The final cold irrigant used was either saline in most of the studies [43,44,45,46,47,48,49,50,51,52,55,57] or sodium hypochlorite (NaOCl) [53,54,56,58]. The temperature of the final irrigant ranged from 1.5 °C to 4 °C and the volume was 5 mL [44,51,58], 10 mL [49,50,51], or 20 mL [43,46,48,52,53,54,55,56] for durations of 1 min [45,51,58], 2 min [47], or 5 min [43,44,46,48,49,52,53,54,55,56]. Two studies did not mention the duration [50,57]. EndoVac negative irrigation was used along with needle syringe activation in five studies [43,44,48,51,57]. Thirteen studies (*n* = 1219) compared the intracanal cryotherapy and control groups in terms of using analgesics postoperatively [43,44,45,46,48,50,51,52,53,54,55,57,58]. None of the patients in either group used analgesics in four studies [43,44,45,57] and fewer patients used analgesics in the intracanal cryotherapy group in eight studies [46,48,50,51,52,53,54,58]. The same number of patients in both groups used analgesics in one study [55]. Three studies did not mention any information about the use of analgesics postoperatively in either group [47,49,56]. Nine studies mentioned that root canal treatment was completed in one visit [43,44,46,49,50,51,52,53,57], six studies performed root canal treatment in two visits [45,47,48,54,55,56], and one study did not mention any information about the number of root canal treatment visits [58].The risk of bias (RoB 2) assessment for the included studies is summarized in Figure 2. According to the RoB 2 [65] assessment for the included sixteen studies, four studies were considered to have a high bias due to “deviations from the intended intervention” [47,50,54,57], two studies were considered to have a high bias due to “missing outcome data” [47,50], and five studies were considered to have a high bias for “measurement of the outcome” [45,50,54,55,57].

### 3.1. Postoperative Pain at 6 h

Six studies (*n* = 450) revealed postoperative pain at 6 h [45,48,49,53,54,57]. A meta-analysis demonstrated a significant statistical diminution in postoperative pain in the intracanal cold therapy group (experimental group) compared with the room temperature irrigant group (control group) (mean difference (MD) = −1.11; 95% confidence interval (CI) = −1.72 to −0.5; *p* = 0.0004; I^2^ = 72%) (Figure 3). The GRADE was low because of concerns with the RoB 2 and the substantial heterogeneity, which indicated that “we have little confidence in the effect estimate and the true effect might be markedly different from the estimated effect” (Table 2).

### 3.2. Postoperative Pain at 24 h

Nine studies (*n* = 763) revealed postoperative pain at 24 h [45,46,48,49,51,53,54,57,58]. A meta-analysis demonstrated a significant statistical diminution in postoperative pain in the intracanal cold therapy group (experimental group) compared with the room temperature irrigant group (control group) (MD = −1.08; 95% CI = −1.79 to −0.38; *p* = 0.003; I^2^ = 92%) (Figure 4). The GRADE was low because of concerns with the RoB 2 and the substantial heterogeneity, which indicated that “we have little confidence in the effect estimate and the true effect might be markedly different from the estimated effect” (Table 2).

### 3.3. Postoperative Pain at 48 h

Five studies (*n* = 440) revealed postoperative pain at 48 h [45,49,51,57,58]. A meta-analysis demonstrated a significant statistical diminution in postoperative pain in the intracanal cold therapy group (experimental group) compared with the room temperature irrigant group (control group) (MD = −0.38; 95% CI = −0.73 to −0.02; *p* = 0.04; I^2^ = 81%) (Figure 5). The GRADE was low because of concerns with the RoB 2 and the substantial heterogeneity, which indicated that “we have little confidence in the effect estimate and the true effect might be markedly different from the estimated effect” (Table 2).

### 3.4. Postoperative Pain at 72 h

Five studies (*n* = 563) revealed postoperative pain at 72 h [46,48,51,53,58]. However, one study had zero scores for the mean and standard deviation for the intracanal cold therapy group (experimental group) [53]; therefore, the MD and CI were not estimable and they were not included in the statistics. A meta-analysis demonstrated a significant statistical diminution in postoperative pain in the intracanal cold therapy group (experimental group) compared with the room temperature irrigant group (control group) (MD = −0.69; 95% CI = −1.34 to −0.05; *p* = 0.04; I^2^ = 90%) (Figure 6). The GRADE was low because of concerns with the RoB 2 and the substantial heterogeneity, which indicated that “we have little confidence in the effect estimate and the true effect might be markedly different from the estimated effect” (Table 2).

### 3.5. Postoperative Pain at 7 Days

Three studies (*n* = 113) revealed postoperative pain at seven days [46,57,58]. Two of these studies [57,58] had zero scores for the mean and standard deviation for the experimental group. Therefore, the MD and CI were not estimable.

## 4. Discussion

The present systematic review and meta-analysis showed that the application of intracanal cryotherapy minimized endodontic pain at 6, 24, 48, and 72 h postoperatively. These results were consistent with previous systematic reviews [59,60]. However, Monteiro et al. in their systematic review and meta-analysis showed that pain was only minimized 6 and 24 h postoperatively [61]. After seven days, there was a reduction in the postoperative pain scores in the intracanal cryotherapy group (experimental group) compared with the room temperature group (control group) but this reduction could not be proven statistically due to inestimable MD and CI values. Our results were in contrast with those reported by Gupta et al. in their systematic review and meta-analysis where they showed that intracanal cryotherapy did not play a significant role in minimizing postendodontic pain [62]. Pain reduction after cryotherapy application is due to several mechanisms including changes in the nerve conduction velocity, inhibition of nociceptors, and a reduction in the metabolic enzyme activity level [67,68,69]. It has been reported that at a 7 °C body temperature, myelinated A-δ fibers are completely deactivated; nonmyelinated C-fibers are deactivated at 3 °C [70,71]. Moreover, Vera et al. in their in vitro study concluded that finalizing the irrigation with a 2.5 °C saline solution decreased the temperature of the external root surface by more than 10 °C for a 4 min period. This decrease in temperature is sufficient to decelerate the inflammatory reaction and reduce the induction of pain-producing substances, leading to local anti-inflammatory effects in the periradicular tissues [36]. Eight of the thirteen studies that compared the use of analgesics between the intracanal cold therapy and the control groups postoperatively showed that fewer patients used analgesics in the intracanal cold therapy group than in the control group. A previous systematic review analyzed the effect of cold therapy on pain reduction and analgesic use after a total knee arthroplasty. Very low certainty evidence was found that cold therapy decreased analgesic use and no evidence showed that it reduced pain [72]. On the other hand, Watkins et al. concluded that cryotherapy decreases postoperative pain and the use of analgesics by patients undergoing major abdominal operations [24].

Our included studies had different numbers of treatment visits to perform root canal treatment; nine studies mentioned that root canal treatment was completed in one visit [43,44,46,49,50,51,52,53,57], six studies performed root canal treatment in two visits [45,47,48,54,55,56], and one study did not mention any information about the number of root canal treatment visits [58]. However, all of them showed a reduction in postoperative pain compared with preoperative pain. This is consistent with a previous systematic review that evaluated the predictors of postoperative endodontic pain and concluded that the number of treatment visits had no significant effect on postoperative pain [73]. On the other hand, Izadpanah et al. in their systematic review and meta-analysis concluded that single-visit root canal therapy has a higher risk of postoperative pain than multiple visits with acceptable statistical heterogeneity and a moderate quality of the studies [74].

The temperature of the final cold irrigant ranged from 1.5 °C to 4 °C for 1 to 5 min among the included studies. The optimal temperature application duration of intracanal cold therapy has not been concluded as none of the studies compared the durations; however, a continuous exposure to a low temperature below −20 °C leads to cell death and tissue destruction [75,76] and intermittent applications of cryotherapy can enhance its therapeutic effect in relieving pain after an acute soft tissue injury [77].

An EndoVac negative pressure irrigation system was used along with needle syringe activation in five studies [43,44,48,51,57]. A previous systematic review analyzed the effect of using an EndoVac versus a needle syringe in controlling postoperative pain; no statistically significant difference was found [60]. On the other hand, it was shown that an EndoVac might cause less postoperative pain due to less apical extrusion of debris [78].

To the best of our knowledge, this is the first systematic review that determined the effect of intracanal cold therapy on postoperative endodontic pain by utilizing the second version of the Cochrane risk of bias tool for randomized trials (RoB 2). The Cochrane risk of bias tool 2 (RoB 2) is designed to focus on the results, leading to a better quality of the risk of bias assessments. Furthermore, the RoB 2 has an approach that applies a granular structure of knowledge by indicating questions and a wider range of possible answers that guide the review authors to focus on the context of clinical trials [79].

In our systematic review, a few concerns of bias arose from the randomization process due to a lack of information about the randomization and the concealment of the allocation process [45,46,49,50,54,57]. A bias associated with inadequate allocation concealment may cause an exaggeration of the estimated treatment effect and affect the meta-analysis results [80]. A bias due to deviations from the intended intervention was observed in eight studies where the people delivering the interventions were not blinded and no information was provided regarding the blinding of the participants [43,45,46,47,50,51,54,57]. A previous study found that the risk of bias in blinding the participants and personnel in endodontics clinical trials is the highest and will lead to an overestimation of the results [81]. Missing outcome data were not clear in two studies [47,50], which might lead to doubts regarding the estimate of the effect [82]. The outcome of postoperative pain was assessed by the patients themselves and no information was provided regarding the awareness of the outcome assessors of the intervention they received in five studies [45,50,54,55,57]. That might lead to bias in the outcome measurement. A bias in the selection of the reported result was found in three studies [50,54,57], which might cause misleading results. These biases might explain the significant reduction in postoperative endodontic pain at 6, 24, 48, and 72 h that was found in the meta-analysis.

The GRADE was utilized to rate the certainty of evidence and it was found to be low due to serious concerns about the risk of bias and inconsistency. These results were in contrast to a previous systematic review that also evaluated the efficacy of intracanal cold therapy for the management of postoperative endodontic pain and the certainty of evidence was judged as moderate [60]. Another systematic review evaluated the same topic but the certainty of evidence was considered very low due to serious concerns with the risk of bias and inconsistency as well as very serious concerns with the imprecision [61].

This systematic review has a few limitations including the variability among the included studies in the study design, the diagnosis of pulpal and periapical areas, the type of teeth included in each study, the preoperative pain status, the experimental group irrigation protocol, the sample size, and the number of treatment visits. These variations among the included studies might affect the intervention effects.

Although intracanal cryotherapy is a simple and inexpensive method that might reduce postoperative endodontic pain, the certainty of evidence illustrated in this research was low. This signifies the need for well-designed trials with precise parameters and variable controls to establish its effective and definitive use in endodontic clinical practice.

## 5. Conclusions

Within the limitations of this study, the application of intracanal cold irrigation showed low certainty of evidence in reducing postoperative endodontic pain. Additional better designed clinical trials are required to establish the effective use of intracanal cryotherapy in controlling postoperative endodontic pain in clinical practice.

## Figures and Tables

**Figure 1 ijerph-18-11750-f001:**
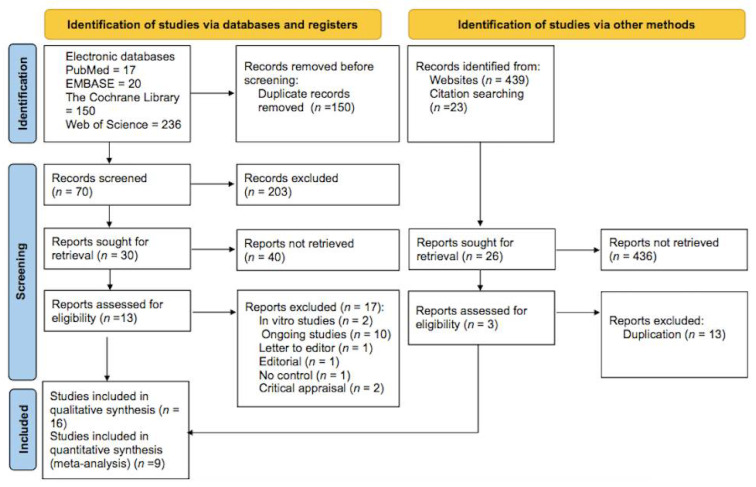
PRISMA 2020 flow diagram for the systematic reviews and meta-analysis.

**Figure 2 ijerph-18-11750-f002:**
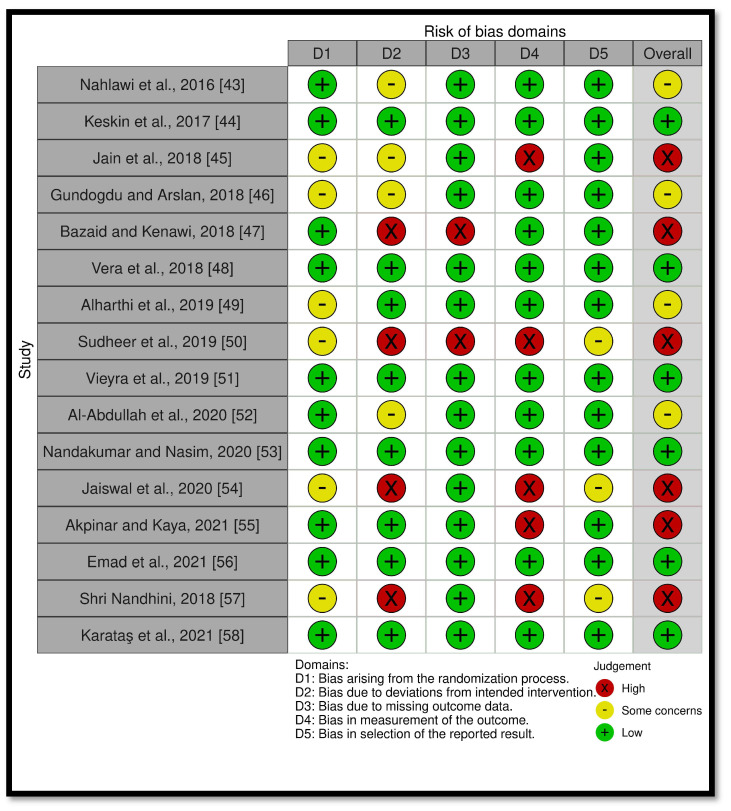
Summary of the risk assessment bias (RoB 2) of the included randomized controlled trials.

**Figure 3 ijerph-18-11750-f003:**
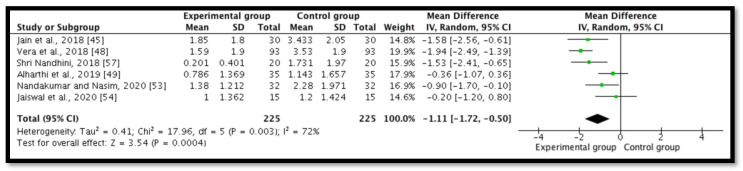
Forest plot comparing postoperative pain between the experimental group (intracanal cryotherapy) and the control group (room temperature) at 6 h. IV: Intravitreal.

**Figure 4 ijerph-18-11750-f004:**
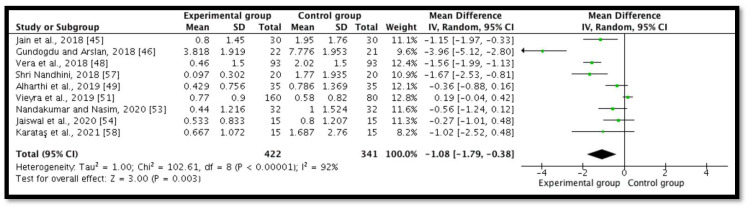
Forest plot comparing postoperative pain between the experimental group (intracanal cryotherapy) and the control group (room temperature) at 24 h.

**Figure 5 ijerph-18-11750-f005:**
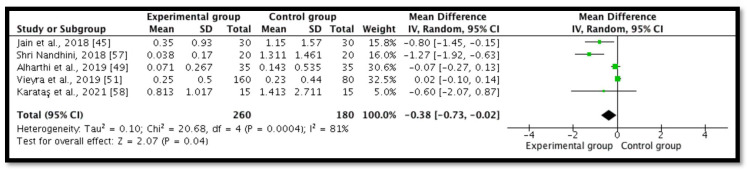
Forest plot comparing postoperative pain between the experimental group (intracanal cryotherapy) and the control group (room temperature) at 48 h.

**Figure 6 ijerph-18-11750-f006:**
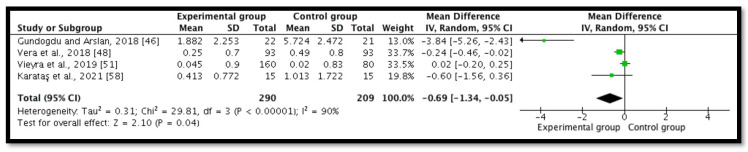
Forest plot comparing postoperative pain between the experimental group (intracanal cryotherapy) and the control group (room temperature) at 72 h.

**Table 1 ijerph-18-11750-t001:** Summary of the main characteristics of the included studies.

Author and Year	Sample Size	Age	Pulpal and Periapical Diagnosis	Type of Teeth	Pain Assessment Tool	Mean and SD of Preoperative Pain in the Cryotherapy Group	Mean and SD of Preoperative Pain in the Control Group	Final Irrigation in the Cryotherapy Group	Final Irrigation in the Control Group	Pain Assessment Times	Treat ment Visits	Conclusion
Nahlawi et al., 2016 ^R^ [43]	75	20–46	Irreversible pulpitis	Single-rooted single canal teeth	VAS (0–10 cm)	2.0 ± 0.9	2.5 ± 0.78	20 mL of 2–4 °C saline for 5 min with needle syringe + EndoVac negative irrigation	20 mL room temperature saline for 5 min with needle syringe + EndoVac negative irrigation	6, 12, 24, 48 h, and 7 days	Single visit	Intracanal cryotherapy eliminated postendodontic pain significantly compared with the control group at 6, 12, 24, and 48 h postoperatively but no significant difference after 1 week
Keskin et al., 2017 ^R^ [44]	170	19–63	Irreversible pulpitis with normal apical tissues or symptomatic apical periodontitis	Maxillary or mandibular incisors/premolars/molars	VAS (0–10 cm)	2.3 ± 0.8	2.0 ± 0.6	5 mL of 2.5 °C saline for 5 min + needle syringe	5 mL room temperature saline for 5 min + needle syringe	24 and 48 h	Single visit	Cryotherapy reduced postoperative pain significantly after 24 h compared with the control group but no significant difference after 48 h
Jain et al., 2018 [45]	60	18–25	Symptomatic irreversible pulpitis with normal apical tissues or apical periodontitis	Mandibular 1st molars	VAS (0–10 cm)	8.12 ± 0.67	8.52 ± 0.75	2.5 °C saline for 1 min + needle syringe	Room temperature saline for 1 min + needle syringe	6, 24, and 48 h	Multi visits	Intracanal cryotherapy was effective in significantly reducing pain during and after endodontic treatment compared with the control group in symptomatic irreversible pulpitis with symptomatic apical periodontitis at 6, 24, and 48 h
Gundogdu and Arslan, 2018 ^R^ [46]	84	≥18	Symptomatic irreversible pulpitis with symptomatic apical periodontitis	Maxillary or mandibular molars	VAS (0–100 cm)	91.32 ± 8.214	94.86 ± 4.293	20 mL of 2.5 °C saline for 5 min + needle syringe	20 mL room temperature saline for 5 min + needle syringe	24, 72 h, 5, and 7 days	Single visit	Intracanal cryotherapy reduced postoperative pain levels and reduced the VAS scores of pain on percussion compared with the levels in the control group after 24 h, 3, 5, and 7 days
Bazaid and Kenawi, 2018 [47]	36	18–40	Irreversible pulpitis with normal apical tissues or apical periodontitis	Molars and premolars	VAS (0–10 cm)	NR	NR	2.5 °C saline for 2 min + side-vented needle	Room temperature saline for 2 min + side-vented needle	24 and 48 h	Multi visits	Using cold normal saline reduced the postoperative pain degree in patients with irreversible pulpitis with apical periodontitis but it did not affect patients with irreversible pulpitis without apical periodontitis after 24 and 48 h postoperatively
Vera et al., 2018 ^R^ [48]	210	18–60	Necrotic pulp with symptomatic apical periodontitis	Single canal teeth	VAS (0–10 cm)	8.847 ± 0.769	8.96 ± 0.789	20 mL of 2.5 °C saline for 5 min	20 mL room temperature saline for 5 min	6, 24, and 72 h	Multi visits	Intracanal cryotherapy reduced the incidence of postoperative pain and the need for medication in patients with necrotic pulp and symptomatic apical periodontitis after 6, 24, and 72 h compared with the control group
Nandhini, 2018 [57]	40	20–50	Symptomatic irreversible pulpitis with normal apical tissues or apical periodontitis	Mandibular premolars	Heft–Parker scale (0–170 mm)	80.35 ± 52.652	85.2 ± 47.32	Cold saline at 2.5 °C + EndoVac microcannula	Saline at room temperature + EndoVac microcannula	6, 12, 24, 48 h, 4, and 7 days	Single visit	Comparison of the control and experimental group values showed significant differences at 6, 12, 24, 48 h, and 4 days and without any significant difference at the end of the 7th day. Use of normal saline as an irrigant reduced the pain intensity slowly whereas the use of cold saline totally abolished pain by the end of the 4th day
Alharthi et al., 2019 ^R^ [49]	105	18–50	Asymptomatic irreversible pulpitis with normal apical tissues	Single canal teeth	VAS (0–10 cm)	0 ± 0	0 ± 0	10 mL of cold (1.5–2.5 °C) saline for 5 min + side-vented needle	10 mL room temperature saline for 5 min + side-vented needle	6, 24, and 48 h	Single visit	Room temperature saline showed comparable results to intracanal cryotherapy in pain reduction at 6, 24, and 48 h postoperatively
Sudheer et al., 2019 [50]	60	18–45	Symptomatic irreversible pulpitis	Single-rooted teeth	VAS (0–10 cm)	NR	NR	10 mL of 2.5 °C saline	10 mL room temperature saline	6, 24, and 48 h	Single visit	Intracanal cryotherapy significantly reduced postoperative pain compared with the control group at 6, 24, and 48 h postoperatively
Vieyra et al., 2019 ^R^ [51]	240	18–65	Vital pulp	Maxillary or mandibular molars, premolar, anteriors	VAS (0–10 cm)	0 ± 0	0 ± 0	10 mL of cold (4 °C + 2.5 °C) saline with cold (4 °C + 2.5 °C) EndoVac microcannula for 1 min	10 mL room temperature saline + EndoVac microcannula for 1 min	24, 48, and 72 h	Single visit	No statistically significant difference between the cold and room temperature saline was found regarding the degree or duration of pain at 24, 48, and 72 h postoperatively
Al-Abdullah et al., 2020 [52]	60	>20	Irreversible pulpitis	Single-rooted with single canal teeth	VAS (0–10 cm)	NR	NR	20 mL of 2–4 °C cold saline for 5 min + needle syringe	20 mL room temperature saline for 5 min + needle syringe	6, 12, 24, 48 h, and 7 days	Single visit	Intracanal cryotherapy eliminated postendodontic pain compared with the control group at 6, 12, 24, and 48 h but the difference was not significant after 7 days
Nandakumar and Nasim, 2020 [53]	64	18–70	Symptomatic irreversible pulpitis with apical periodontitis	Molars and premolars	VAS (0–10 cm)	4.72 ± 1.373	4.66 ± 1.096	20 mL of 2–4 °C NaOCl for 5 min + needle syringe	20 mL room temperature NaOCl for 5 min + needle syringe	6, 24, and 48 h	Single visit	Intracanal cryotherapy significantly reduced analgesic consumption and postoperative pain at 6, 24, and 48 h compared with the control group
Jaiswal et al., 2020 [54]	30	NR	Necrotic pulp with symptomatic apical periodontitis	NR	VAS (0–10 cm)	NR	NR	20 mL of 2.5 °C saline for 5 min + needle syringe	20 mL room temperature saline for 5 min + needle syringe	6 and 24 h	Multi visits	No statistically significant difference between the cold and room temperature saline in reducing postoperative pain and analgesic consumption at 6 and 24 h
Akpinar and Kaya, 2021 ^R^ [55]	94	18–65	Irreversible pulpitis with symptomatic apical periodontitis	Mandibular molars	VAS (0–100 cm)	NR	NR	20 mL of 2.5 °C saline for 5 min + 30 gauge side-hole special irrigation tip	5 mL of distilled water + 30 gauge side-hole special irrigation tip	4, 8, 12, 24, 48, and 72 h	Multi visits	Intracanal cryotherapy significantly reduced postoperative pain compared with the control group at 4, 8, 12, 24, 48, and 72 h
Emad et al., 2021 [56]	48	20–50	Symptomatic apical periodontitis	Single-rooted teeth	VAS (0–10 cm)	NR	NR	20 mL of 5% 2.5 °C NaOCl for 5 min + side-vented (30G) needle	20 mL of 5% room temperature NaOCl for 5 min + side-vented (30G) needle	12, 24, 48, 72 h, and 7 days	Multi visits	The cryotherapy irrigation protocol showed lower levels of postoperative pain compared with the control group at 12, 24, 48, 72 h, and 7 days
Karataş et al., 2021 ^R^ [58]	45	Mean = 27	Necrotic pulp with asymptomatic apical periodontitis	Incisors, canines, premolars	VAS (0–100 cm)	NR	NR	5 mL of 1% 2 °C NaOCl for 1 min + syringe	5 mL of 1% 25 °C NaOCl for 1 min + syringe	24, 48, 72 h, 5, and 7 days	NR	No statistically significant difference between the cold and room temperature NaOCl in postoperative pain reduction and in postoperative analgesic intake

^R^ Registered in PubMed. VAS: Visual analogue scale; NR: Not recorded.

**Table 2 ijerph-18-11750-t002:** Certainty of evidence. Question: Is intracanal cryotherapy effective in reducing postoperative endodontic pain compared with a room temperature irrigant (Control)?; Setting: A dental clinic; Participants: Adult patients (≥18 years old) with permanent teeth that have pulpal or periradicular pathosis; Intervention: Final irrigation with a cold irrigant; Comparator: Final irrigation with an irrigant at room temperature.

Certainty Assessment							Number of Patients		Effect	Certainty ^§^
Number of Studies	Study Design	Risk of Bias	Inconsistency	Indirectness	Imprecision	Other Considerations	Intracanal Cryotherapy	Room Temperature Irrigant (Control)	Mean Difference (95% CI *)	
**Postoperative pain at 6 h**										
6	Randomized clinical trials	Serious ^a^	Serious ^b^	Not serious ^c^	Not serious ^d^	None	225	225	−1.11 (−1.72 to −0.5)	⨁⨁◯◯ LOW
**Postoperative pain at 24 h**										
9	Randomized clinical trials	Serious ^a^	Serious ^b^	Not serious ^c^	Not serious ^d^	None	422	341	−1.08 (−1.79 to −0.38)	⨁⨁◯◯ LOW
**Postoperative pain at 48 h**										
5	Randomized clinical trials	Serious ^a^	Serious ^b^	Not serious ^c^	Not serious ^d^	None	260	180	−0.38 (−0.73 to −0.02)	⨁⨁◯◯ LOW
**Postoperative pain at 72 h**										
5	Randomized clinical trials	Serious ^a^	Serious ^b^	Not serious ^c^	Not serious ^d^	None	290	209	−0.69 (−1.34 to −0.05)	⨁⨁◯◯ LOW

* CI: Confidence interval; ^§^ GRADE (Grading of Recommendations, Assessment, Development and Evaluation) certainty ratings: very low certainty: the authors have very little confidence in the effect estimate (the true effect is probably markedly different from the estimated effect); low certainty: the authors have little confidence in the effect estimate (the true effect might be markedly different from the estimated effect); moderate certainty: the authors have moderate confidence in the effect estimate (the authors believe that the true effect is probably close to the estimated effect); high certainty: the authors have a lot of confidence that the true effect is similar to the estimated effect; ^a^ concerns of a risk of bias; ^b^ substantial heterogeneity; ^c^ direct comparison; ^d^ narrow confidence interval.

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
