# Peer review of "Is Intracanal Cryotherapy Effective in Reducing Postoperative Endodontic Pain? An Updated Systematic Review and Meta-Analysis of Randomized Clinical Trials"

_ijerph, 2021, doi:10.3390/ijerph182211750_

Round 1

Reviewer 1 Report

Thank you very much for the opportunity to review this manuscript. I think this systematic review lacks a higher amount of studies. There are only 16 qualitative studies analysis, and the quantitative analysis contained 9 studies. Therefore, the conclusion is evident and there is a low-quality evidence in the systematic review. The manuscript is well organized and described but it does not add any novelty to the scientific literatue and it is not going to change any clinical practice.

Reviewer 2 Report

Dear author, thank you for your manuscript submission. The paper is very well written, the study is well conducted and  accurate. However some minor very easy to correct modifications are suggested before publication.

Consider adding the R of registered in Pubmed, etc...

Line 19 : Mean difference ( MD) means standard deviation?

I would clarify  the groups distribution in the abstract t ( test : cold) control ( no cold applications)

CI in the abstract should be remove, adding the CI in the Materials and methods is enough

verify that line 43-45 is correct and re-redact again for easy understanding

In the PICO question  the objective is to determine the pain  from day ONE, consequently the papers that  include results before the day one should be not included and not mentioned in the conclusions, this point should be corrected.

Line 89 ( ad the initials of the reviewers) idem 93

Line 94 ( Excel, Microsoft Office (R), ,......, all the materials used should include company, location... etc)

Figure 1, remove the red lines from the image and add the R for registered

Inclusion/exclusion criteria should be more clearly defined

References in line 116-117 should be written in the table rather Than the text

Is there any information in the reviewed papers about  the application time and the temperature of the solution?

Congratulations again for the manuscript

Reviewer 3 Report

The present article project title is “Is intracanal cryotherapy effective in reducing postoperative endodontic pains? An updated systematic review and meta-analysis of randomized clinical trials.” There are some weaknesses through the manuscript which need improvement. Therefore, the submitted manuscript cannot be accepted for publication in this form. My comments and suggestions are as follows:

  1. In the abstract section: lines 23 – 25, pg 1. If this could be accepted as a study conclusion, please do not compare it with Gupta study (lines 120-122, pg13), since as authors cited “low-quality evidence suggests (…)”
  2. Line 94, pg 3. Why do authors not included the journal publication quartile and deduce the final conclusions accordingly to recognize publication score?
  3. Lines 16/17, pg 9. Please correct the references.
  4. Line 26, pg 9. Six instead of 6.
  5. Lines 31/36, pg 9. I do not agree with the final quality outcomes as well as described in Figure 2. Please re-evaluate/review the studies accordingly with the referenced guidelines.
  6. If authors do have a systematic review and meta-analysis why the conclusion is only that? Please introduce novelty and curiosity within this review. Moreover, the meta-analysis state that at 24, 48 and 72h “a significant statistical diminution in postoperative pain in the intracanal cold therapy group (experimental group) compared to the room temperature irrigant group (control group)” even with a low GRADE. But why the conclusion was not about this result as an example? This is not mentioned in the discussion section.
  7. Conclusion: the section has 2 sentences. The first is a good concept described in a poor sentence. The second phrase must be cited in terms of “higher quality” the outcomes of the present clinical randomized controlled clinical trials with the best outcomes.

Round 2

Reviewer 3 Report

I do not agree with the final conclusion text. Please rephrase it. 
